# Mpox knowledge and positive attitudes in Sub-Saharan African healthcare workers after 2022 outbreak of disease: A systematic review and meta-analysis

Melaku Laikemariam[1]*, Alemayehu Molla Wollie[2], Amare Mebrat Delie[3], Abebe Yenesew[4], Abateneh Melkamu[5], Habtamu Ayele[6], Yihalem Abeje[1]

1 Department of Midwifery, College of Medicine and Health Sciences, Injibara University, Injibara, Ethiopia, 2 Department of Psychiatry, College of Medicine and Health Sciences, Injibara University, Injibara, Ethiopia, 3 Department of Public Health, College of Medicine and Health Sciences, Injibara University, Injibara, Ethiopia, 4 Department of Medical Laboratory Science, College of Medicine and Health Sciences, Injibara University, Injibara, Ethiopia, 5 Department of Medical Laboratory Science, College of Health Sciences, Debre Markos University, Debre Markos, Ethiopia, 6 Department of Midwifery, College of Health Sciences, Debre Markos University, Debre Markos, Ethiopia

* laikemariam2014@gmail.com

## Abstract

### Background

Mpox remains a public health emergency of international concern, especially in regions beyond its usual endemic areas in Africa. Assessing healthcare workers good knowledge and positive attitudes is essential for effective prevention and control efforts. This systematic review and meta-analysis aim to determine the pooled good knowledge and positive attitudes toward mpox among healthcare workers in Sub-Saharan Africa after the 2022 outbreak.

### Methods

We searched major databases for relevant studies published up to June 25, 2025. Studies reporting knowledge and/or attitudes toward mpox were included. Study quality was assessed using a standardized appraisal tool, and heterogeneity was assessed using the $I^2$ statistic. Data were extracted using a standardized protocol, and a random-effects model was used to calculate pooled prevalence estimates with 95% confidence intervals.

### Results

The meta-analysis included sixteen and eight studies in knowledge and attitude analyses, respectively, to estimate the pooled prevalence. The pooled prevalence of good knowledge and positive attitudes toward monkeypox was 45.3% (95% CI: 36.8, 53.9) and 53.8% (95% CI: 43.0, 64.7), respectively. Significant heterogeneity was observed

**Data availability statement:** All data underlying the findings are available within the manuscript and its supplementary files. The characteristics and extracted data from included studies are presented in Table 2. Additional data, including the PRISMA checklist, search strategy, included and excluded study, data extraction, and missing data handling, are provided in S1-S5 Files.

**Funding:** The author(s) received no specific funding for this work.

**Competing interests:** The authors have declared that no competing interests exist.

across studies; however, both statistical tests (Egger's test, p = 0.14; Begg's test, p = 0.19) indicated no significant publication bias.

## Conclusion

The good knowledge and positive attitudes of healthcare workers toward mpox were low and unsatisfactory in sub-Saharan Africa. The review result underscores the need for targeted interventions to improve healthcare providers' understanding of mpox transmission, prevention, and management. Targeted educational programs and training are needed to improve the preparedness of healthcare workers for mpox outbreaks and other emerging diseases.

## Author summary

Mpox still poses a major public health issue in Sub-Saharan Africa. Healthcare workers stand right on the front lines against these outbreaks. Despite their crucial role, there has been no comprehensive regional assessment of their preparedness in terms of knowledge and attitudes in Sub-Saharan Africa. To address this gap and guide training, we conducted a systematic review and meta-analysis of available cross-sectional studies, pooling data from 16 studies involving 6047 healthcare workers for knowledge and 8 studies involving 3303 healthcare workers for attitude across multiple African countries. We confirmed that fewer than half (45.3%) of healthcare workers had a good knowledge of mpox transmission, clinical signs, and prevention measures. On the other hand, more than half (53.8%) had shown positive attitudes toward controlling the disease. These results demonstrated a critical gap in healthcare workers' knowledge and their positive attitudes toward responding to the disease outbreak. To improve mpox knowledge and attitudes of healthcare workers, targeted and focused training programs, especially in high-risk and rural areas, are urgently needed. Improving good knowledge and positive attitudes among healthcare workers will enhance outbreak detection and preparedness efforts across Sub-Saharan Africa.

## Introduction

Mpox, a zoonotic disease endemic to Central and West Africa, poses a growing public health threat due to increased human-to-human transmission [1,2]. Clinically, mpox resembles smallpox and begins with symptoms like fever, headache, and muscle pain [1,3]. Transmission to humans occurs through contact with infected fluids, skin lesions, or mucous membranes [2,4].

The recent global outbreaks prompted the World Health Organization (WHO) to declare mpox a public health emergency of international concern in August 2024 [5], with Africa bearing a disproportionate burden [5,6]. This underscores the global

importance of coordinated prevention and control measures [1,5]. As of June 2024, the World Health Organization (WHO) reported nearly 100,000 confirmed cases and over 200 deaths worldwide, with African nations disproportionately affected [7].

Good knowledge and positive attitudes among healthcare workers (HCWs) is fundamental for effective mpox prevention and control [8–10]. Effective prevention and control of mpox rely heavily on healthcare workers' (HCWs) good knowledge and positive attitudes [11–13]. Previous global systematic reviews and meta-analyses have reported low pooled estimates of good mpox knowledge (26–32%) and positive attitudes (34.6%) among healthcare workers (HCWs) [14,15].

However, these global syntheses may mask critical regional variations, particularly in Sub-Saharan Africa (SSA), where mpox is endemic and healthcare systems face unique structural, resource, and contextual challenges [1,8,9,16]. Furthermore, studies showed poor knowledge and attitudes among healthcare workers (HCWs) regarding mpox, which may hinder effective management and control efforts [14,15,17].

Evidence from individual Sub-Saharan Africa (SSA) countries suggests widely variable and often suboptimal levels of healthcare workers (HCWs) knowledge and attitudes toward mpox. This inconsistency highlights a significant gap; there is no comprehensive, region-specific synthesis to determine the pooled prevalence of mpox good knowledge, and positive attitudes among healthcare workers (HCWs) in Sub-Saharan Africa (SSA) after the major 2022 global outbreak. Understanding this context is crucial, as Sub-Saharan Africa`s (SSA) epidemiological landscape, healthcare infrastructure, and access to information differ substantially from global patterns, necessitating tailored interventions. This review hypothesized that the pooled proportion of healthcare workers with good mpox knowledge and positive attitudes would be less than optimal levels, reflecting the gaps in their awareness and preparedness despite the 2022 outbreak.

Therefore, this systematic review and meta-analysis aim to determine the pooled prevalence of good knowledge and positive attitudes toward mpox among healthcare workers in Sub-Saharan Africa following the 2022 outbreak. By synthesizing existing evidence, this study will provide a clearer picture of the current state of healthcare workers preparedness in the region, identify knowledge-attitude gaps, and inform the development of targeted, context-appropriate educational and training programs to enhance mpox control and public health preparedness in Sub-Saharan Africa (SSA).

## Method and materials

### Ethics statement

Ethical approval was not required for this systematic review and meta-analysis, as it synthesizes data from previously published studies. Majority of primary studies included in this analysis reported obtaining ethical approval from their respective institutional or national boards prior to data collection in accordance with the Declaration of Helsinki, but one primary study (Iwuafor, 2023) is not clearly reported ethical statement for the study.

### Protocol registration

This systematic review and meta-analysis was conducted in strict adherence to the Preferred Reporting Items for Systematic Review and Meta-Analysis (PRISMA) 2020 guidelines [18]. The completed PRISMA checklist is provided in Supplementary file S1 File.

The study protocol was registered in the International Prospective Register of Systematic Reviews (PROSPERO) with registration ID: **CRD4201075654.**

### Information sources and searching strategy

Two authors (ML&YA) independently conducted thorough searches across seven databases, including PubMed/Medline, Scopus, EMBASE, Web of Science, CINAHL, and African Journal Online (AJOL), and Google Scholar. Furthermore, a confirmatory search through Google was performed to ensure no primary studies were missed.

("monkeypox"[MeSH Terms] OR "monkeypox" OR "mpox" OR "orthopoxvirus") AND ("knowledge"[MeSH Terms] OR "knowledge" OR "awareness" OR "understanding" OR "attitude"[MeSH Terms] OR "attitude" OR "perception" OR "practice") AND ("health personnel"[MeSH Terms] OR "healthcare worker" OR "health care provider" OR "nurse" OR "physician" OR "doctor" OR "clinician") AND "Sub-Saharan Africa" OR "Africa" OR "Ethiopia" OR "Nigeria" OR "Kenya" OR "Uganda" OR "Cameroon" OR "Ghana" OR "Tanzania") AND 2022–2025. The full electronic search strategies for each database were provided in the supplementary file S2 File. The reference lists of all identified articles were checked for additional sources.

### Eligibility criteria

All observational studies reporting knowledge, attitude, or both, and published in English up to June 25, 2025, were included in the meta-analysis. Conversely, studies conducted on non-healthcare worker populations, qualitative studies, reviews, case reports, and studies that did not report the outcomes of interest (for attitude analysis) or that were unrelated to our study objective were excluded from the meta-analysis. The included and excluded studies were described in the supplementary file with reason of exclusion S3 File.

### Outcome of interest

This meta-analysis assessed the pooled prevalence of good knowledge and positive attitudes toward mpox among healthcare workers in Sub-Saharan Africa.

**Good knowledge:** was defined based on the reporting of adequate understanding of mpox transmission, symptoms, prevention, diagnosis, and treatment, as referenced from included studies [15].

**Positive attitudes:** were defined as healthcare workers' confidence in mpox control, belief in preventive measures, and perception that their actions could prevent spread [15].

### Data extraction

Data were extracted independently by two reviewers (ML & YA) using a standardized Microsoft Excel spreadsheet prepared format to extract relevant data from the included primary studies. The following data were extracted: first author name, publication year, country, study design, sample size (total and included), response rate, number of healthcare workers with good knowledge of mpox, prevalence of good knowledge, prevalence of positive attitudes, and number of healthcare workers with positive attitudes toward mpox. Finally, reviewers independently verified the extracted data to ensure accuracy and eliminate incorrect information. The full data extraction excel sheet was described in the supplementary file S4 File.

### Study selection and quality assessment

Two reviewers (ML&YA) independently screened records using predefined criteria. Subsequently, the 3rd reviewer (AM) thoroughly reviewed the full texts of potentially eligible studies in the first review to determine if they met the inclusion criteria. Any disagreement between reviewers was solved by discussion, and if not, the third reviewer (AD) was engaged. The PRISMA flow diagram was used to summarize the study selection processes. Methodological quality was assessed using the Joanna Briggs Institute (JBI) critical appraisal checklist for observational studies [19]. All studies met the quality threshold (score ≥5/8), indicating moderate to high methodological quality, and they were included in the review (Table 1).

Each study was assessed across eight domains: sample definition, setting description, exposure measurement, outcome criteria, confounding identification, confounding strategies, outcome measurement, and statistical analysis. Studies scoring ≥5/8 were included, indicating moderate to high methodological quality.

**Table 1. Quality assessment of included studies using the Joanna Briggs Institute (JBI) checklist for observational studies.**

| Study | Inclusion in the sample was clearly defined | Study subjects and the setting are described in detail | Exposure is measured in a valid and reliable way | Objective, standard criteria for measurement of the condition? | Confounding factors identified | Strategies to deal with confounding factors are stated | Outcomes are measured in a valid and reliable way | Was an appropriate statistical analysis applied? | Total score |
|---|---|---|---|---|---|---|---|---|---|
| Almaw et al [20] | Yes | Yes | No | Yes | Yes | Yes | Yes | Yes | 7 |
| Aynalem et al [21] | Yes | No | Yes | Yes | No | No | Yes | Yes | 5 |
| Beyna et al [22] | Yes | No | No | No | Yes | Yes | Yes | Yes | 7 |
| Fetensa et al [23] | No | Yes | No | Yes | Yes | Yes | Yes | Yes | 6 |
| Kiros et al [24] | Yes | Yes | Yes | Yes | Yes | Yes | Yes | Yes | 8 |
| Sofonias et al [25] | Yes | No | Yes | Yes | No | No | Yes | Yes | 5 |
| Abdulmumin et al [26] | Yes | Yes | Yes | Yes | No | Yes | Yes | Yes | 7 |
| Oche et al [27] | Yes | No | Yes | No | No | Yes | Yes | Yes | 5 |
| Orok et al [28] | No | Yes | Yes | Yes | No | Yes | Yes | Yes | 6 |
| Ajayi et al [29] | Yes | Yes | Yes | Yes | Yes | Yes | Yes | Yes | 8 |
| Uche et al [30] | Yes | Yes | Yes | Yes | No | No | Yes | Yes | 6 |
| Iwuafor et al [31] | Yes | No | Yes | Yes | Yes | Yes | Yes | Yes | 7 |
| Epipode et al [32] | Yes | No | Yes | Yes | Yes | Yes | Yes | Yes | 8 |
| Namakula et al [33] | Yes | Yes | No | No | Yes | No | Yes | Yes | 5 |
| Joyce et al [34] | Yes | Yes | Yes | Yes | No | Yes | Yes | Yes | 7 |
| Nka et al [35] | Yes | Yes | Yes | Yes | Yes | Yes | No | Yes | 7 |

## Data analysis

The result of this meta-analysis was analyzed using STATA version 17. Heterogeneity across primary studies was checked using the standard $I^2$ statistical test. Random effect model was computed to estimate Der Simonian and Laird's pooled prevalence of good knowledge and positive attitudes toward mpox. Statistical tests (Egger's test and Begg's test) were used to assess publication bias. Subgroup analysis was conducted based on country (Ethiopia and Nigeria), because these countries had a sufficient number of published studies (E = 6, N = 6) to support robust subgroup analysis, unlike other Sub-Saharan Africa (SSA) nations with ≤2 studies. Furthermore, to explore potential sources of heterogeneity, random-effects meta-regression was conducted with publication year and sample size as continuous moderator variables, with the proportion of variance explained reported using the $R^2$ statistic. Additionally, sensitivity analysis was conducted to assess the effect of individual study on the pooled prevalence of good knowledge. Subgroup and meta-regression analysis were not conducted for attitude analysis due to the limited number of studies (n < 10 studies). Forest plots were used

to present the results of this meta-analysis. Missing data were handled by exclusion, no imputation was performed. All analyses were conducted using complete case data as reported in the original studies S5 File.

## Result

### Study selection

Initially, 2,562 records were identified from various databases using the "MeSH" search combined with Boolean operators. Following the initial assessment, 2437 were excluded as irrelevant and unrelated to our study objectives, and 90 were excluded due to duplication. Further screening of 35 articles resulted in the exclusion of 19 due to irrelevance to our study. After attempts to retrieve 16, all were retrieved and available. Sixteen articles underwent eligibility assessment; 16 articles were included in the knowledge analysis, but 8 articles were excluded from attitude analysis because they did not report outcomes of interest. The final analyses included 16 articles [20–35] for knowledge and 8 articles [20–24,26,28,34] for attitude meta-analysis (Fig 1).

### Characteristics of included studies

This systematic review included 16 cross-sectional studies on knowledge (n = 6,047; response rate 91.8%) and 8 on attitudes (n = 3,303; response rate 94.9%). Studies were primarily from Ethiopia and Nigeria, published between 2023 and 2025, with sample sizes ranging from 164 [31] to 749 participants [23]. The prevalence of good knowledge ranged from 22.5% to 82.1%, and positive attitudes ranged from 35.8% to 80.0% (Table 2).

### Publication bias assessment

Publication bias was assessed for knowledge studies using Egger's and Begg's tests. Both tests indicated no significant publication bias (Egger's test: $p = 0.14$; Begg's test: $p = 0.19$). A funnel plot showed slight asymmetry but no statistically significant bias (Fig 2). Publication bias was not assessed for attitude studies due to the small number of included studies (n < 10).

### Pooled prevalence of good mpox knowledge among healthcare workers

The pooled prevalence of good knowledge regarding mpox among healthcare workers in Sub-Saharan Africa was 45.3% (95% CI: 36.8, 53.9). Significant heterogeneity ($I^2 = 98.1\%$) was present, reflecting a wide range of variation in study settings. The overall effect test was statistically significant (z = 10.04, p = 0.000) in spite of such heterogeneity. Individual point estimates ranged between 22.5% (17.1, 27.9) [28] and 82.1% (78.4, 85.9) [26] (Fig 3).

### Pooled prevalence of positive attitudes toward mpox among healthcare workers

The pooled prevalence of positive attitudes toward mpox was 53.8% (95% CI: 43.0, 64.7). The individual estimates ranged from 35.8% (32.0, 39.6) [20] to 80.0% (75.7, 84.3) [34]. High heterogeneity ($I^2 = 97.7\%$) suggests that the findings are not consistent across the included studies (Fig 4).

### Subgroup analysis of mpox knowledge by country

Subgroup analysis compared mpox knowledge among healthcare workers in Ethiopia and Nigeria. The pooled prevalence of good knowledge was 44.3% (95% CI: 34.5, 54.2) in Ethiopian studies and 47.6% (95% CI: 27.2, 67.9) in Nigerian studies. No statistically significant difference was observed between the two countries (p = 0.78). High heterogeneity persisted within both subgroups (Ethiopia: $I^2 = 96.1\%$; Nigeria: $I^2 = 99.0\%$), indicating that additional contextual and methodological factors beyond country likely contribute to variability in knowledge estimates (Fig 5).

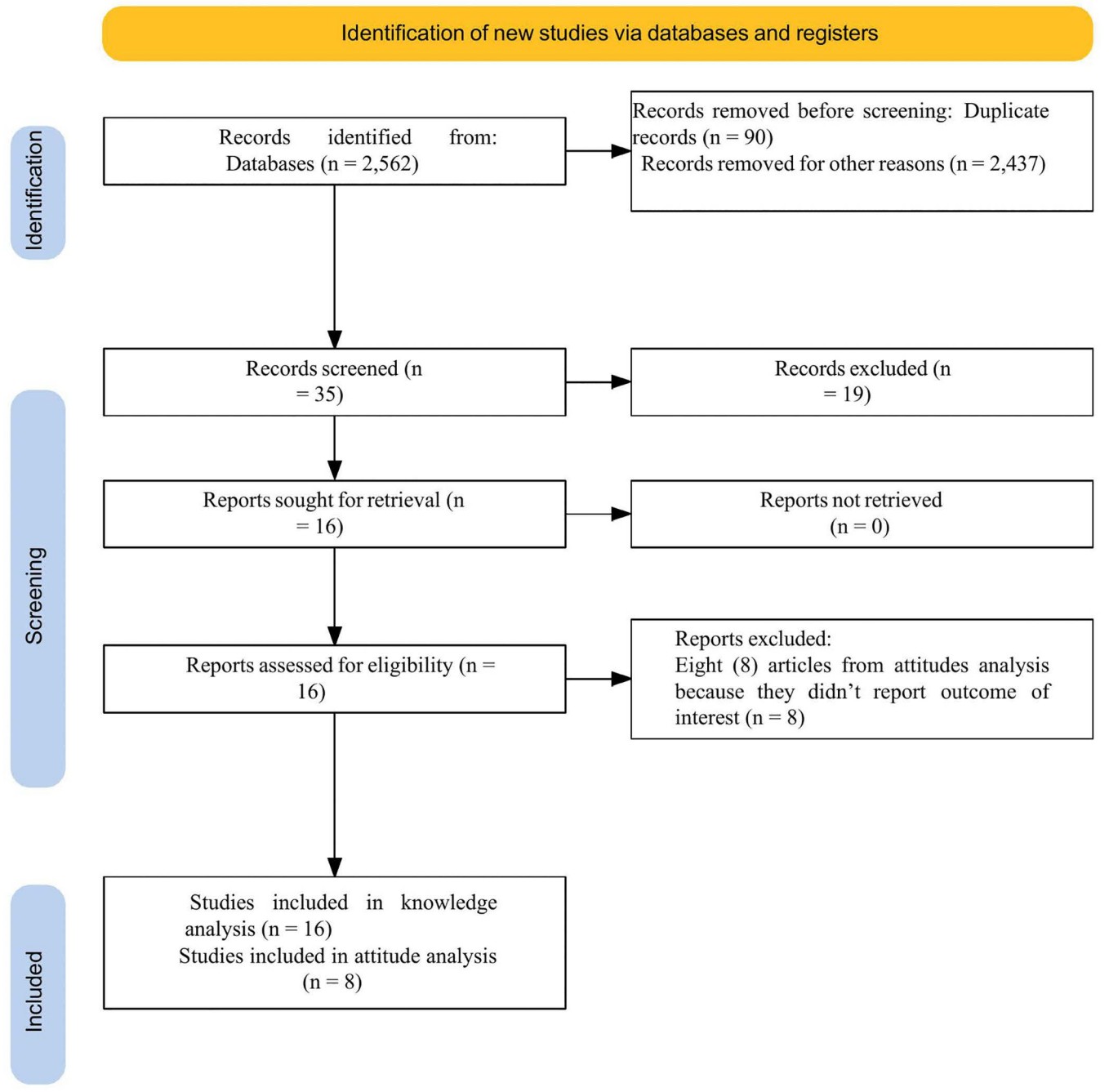

**Fig 1. PRISMA 2020 flow diagram of study identification, screening, and inclusion for the analysis.**

## Meta-regression analysis

Random-effects meta-regression was conducted to explore potential sources of heterogeneity. Publication year was not significantly associated with mpox knowledge prevalence (coefficient = −1.84, 95% CI: −13.5 to 9.9; p = 0.76), nor was sample size (coefficient = 0.04, 95% CI: −0.02 to 0.09; p = 0.19). Both models explained minimal between-study variance

**Table 2. Characteristics of included studies assessing mpox knowledge and attitudes among healthcare workers in Sub-Saharan Africa.**

| Study | Year | Country | Study design | Calculated sample | Included sample | Response rate (%) | Good knowledge N (%) | Positive attitude N (%) |
|---|---|---|---|---|---|---|---|---|
| Almaw et al [20] | 2024 | Ethiopia | Cross sectional | 640 | 620 | 96.9 | 361 (58.3) | 222 (35.8) |
| Aynalem et al [21] | 2025 | Ethiopia | Cross sectional | 201 | 200 | 99.5 | 77 (38.5) | 124 (62) |
| Beyna et al [22] | 2025 | Ethiopia | Cross sectional | 401 | 382 | 95.3 | 185 (48.4) | 188 (49.2) |
| Fetensa et al [23] | 2025 | Ethiopia | Cross sectional | 800 | 749 | 94.0 | 423 (56.5) | 386 (51.5) |
| Kiros et al [24] | 2025 | Ethiopia | Cross sectional | 384 | 384 | 100.0 | 108 (28.1) | 145 (37.8) |
| Sofonias et al [25] | 2024 | Ethiopia | Cross-sectional | 226 | 209 | 92.4 | 70 (35.4) | NR |
| Abdulmumin et al [26] | 2023 | Nigeria | Cross sectional | 402 | 402 | 100.0 | 330 (82.1) | 281 (69.9) |
| Oche et al [27] | 2024 | Nigeria | Cross sectional | 210 | 210 | 100.0 | 151 (72%) | NR |
| Orok et al [28] | 2024 | Nigeria | Cross sectional | 227 | 227 | 100.0 | 51 (22.5) | 101 (44.5) |
| Ajayi et al [29] | 2023 | Nigeria | Cross sectional | 338 | 316 | 93.0 | 107 [33] | NR |
| Uche et al [30] | 2024 | Nigeria | Cross sectional | 609 | 609 | 100.0 | 318 (52.2) | NR |
| Iwuafor et al [31] | 2023 | Nigeria | Cross sectional | 450 | 164 | 36.4 | 38 (23.2) | NR |
| Epipode et al [32] | 2025 | Burundi | Cross sectional | 471 | 471 | 100.0 | 330 (70) | NR |
| Namakula et al [33] | 2025 | Uganda | Cross sectional | 423 | 423 | 100.0 | 186 (44) | NR |
| Joyce et al [34] | 2025 | Uganda | Cross sectional | 426 | 339 | 80.0 | 202 (60) | 270 (80) |
| Nka et al [35] | 2024 | Cameroon | Cross sectional | 377 | 342 | 90.7 | 144 (42.1) | NR |

**N**= Frequency of outcome, **%**: Prevalence of outcome, **NR:** No reported outcome of interest (attitude)

($R^2=0.63\%$ and 2.07%, respectively), indicating that neither temporal trend nor study scale meaningfully accounted for the high heterogeneity observed across studies.

## Sensitivity analysis

A sensitivity test was conducted to detect each study's effect on the overall prevalence of good knowledge by excluding one study at a time. Based on the findings from the sensitivity analysis, no studies in the review impacted the pooled prevalence of good knowledge toward mpox (Fig 6).

## Discussion

This meta-analysis found that the pooled prevalence of good mpox knowledge among Sub-Saharan African healthcare workers was 45.3%, while positive attitudes reached 53.8%, reflecting moderate but insufficient levels of awareness and readiness across the region. The pooled prevalence of good knowledge of this review aligns with the findings reported in Uganda [33].

The pooled knowledge estimate is substantially higher than those reported in pre-2022 global meta-analyses (26.0% and 32.0%) [14,15], likely due to heightened media coverage, updated guidelines, and targeted training initiatives following the 2022 global outbreaks. The most probable reason for this improved good knowledge is that our study included research conducted after the 2022 global outbreaks, whereas the previous meta-analyses incorporated studies carried out before these events, when awareness was poor due to limited information. However, the regional estimate of good knowledge remains lower than several individual studies from non-SSA settings such as Vietnam [36], Saudi Arabia [37], Egypt [38], and Nepal [39], which reported 50.0%, 55.0%, 55.5%, 60.4% respectively, as well as within SSA in Ethiopia [23], Uganda [34], and Nigeria [26], which reported 56.5%, 60.0%, 82.1% respectively. This discrepancy suggests that while post-outbreak awareness has risen globally, SSA continues to face structural and informational barriers, such as variable

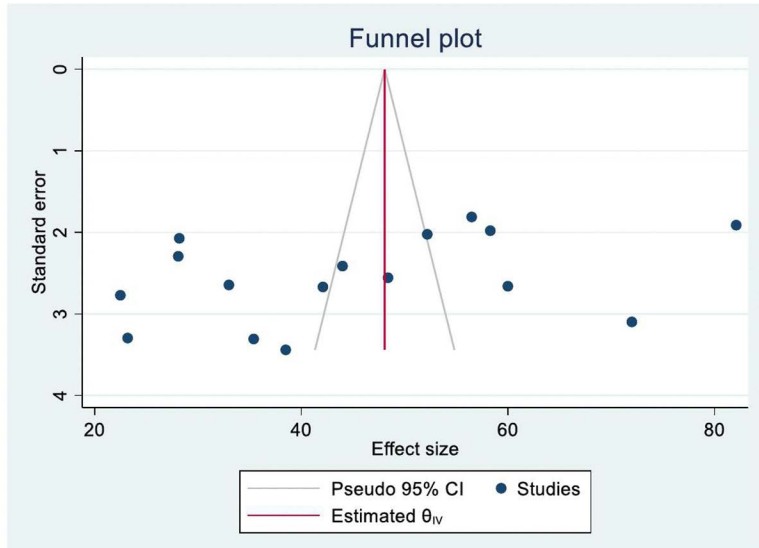

**Fig 2. Funnel plot illustrating assessment of potential publication bias in the mpox knowledge meta-analysis.**

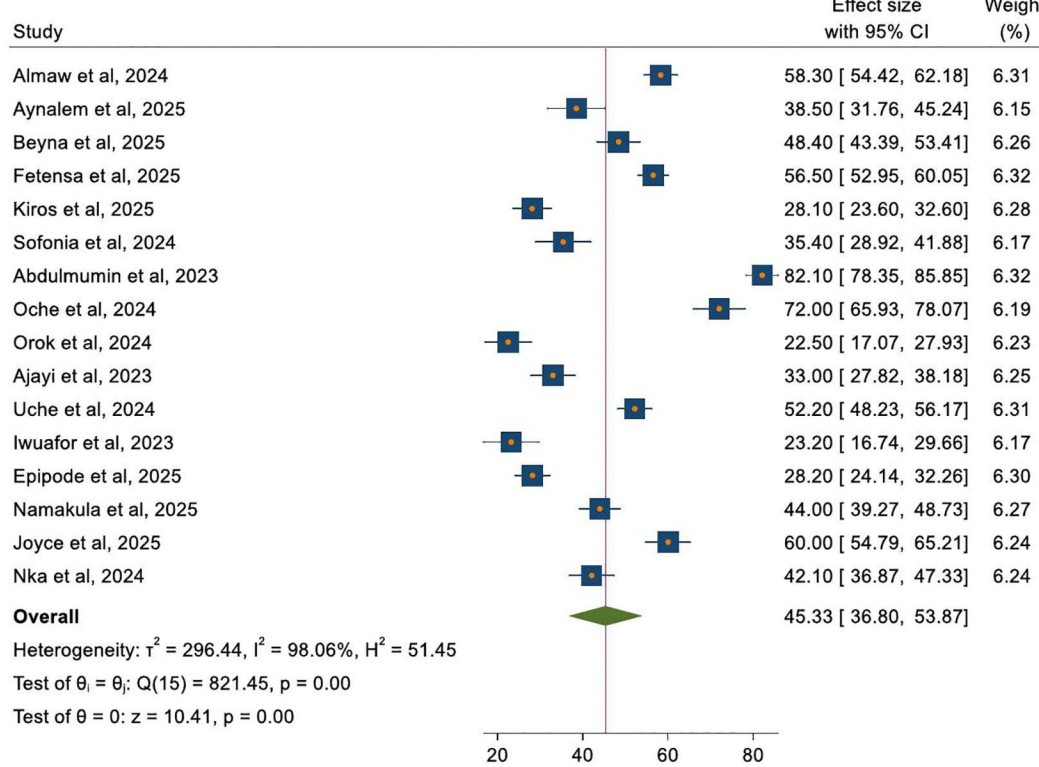

**Fig 3. Forest plot illustrating the pooled prevalence of good mpox knowledge among healthcare workers (16 studies).**

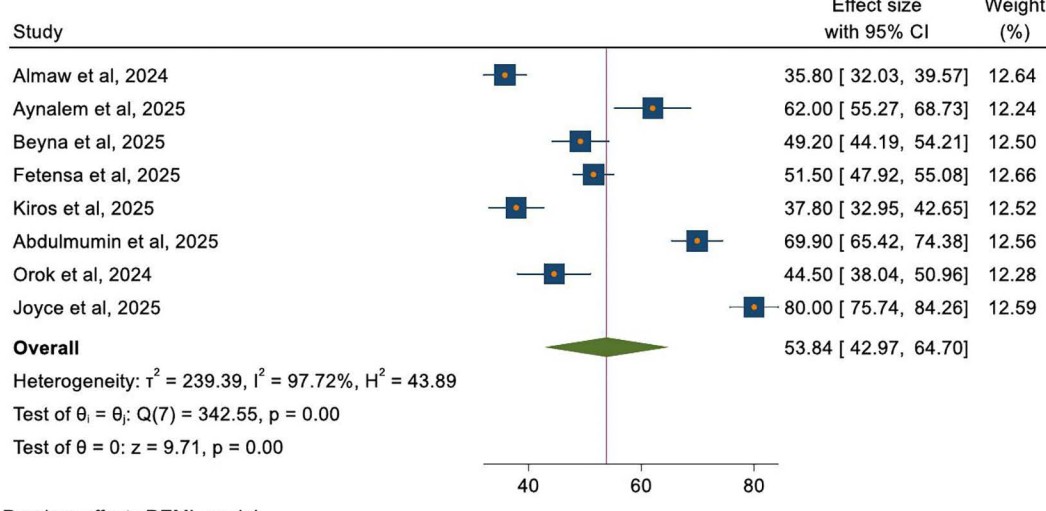

**Fig 4. Forest plot illustrating the pooled prevalence of positive attitudes toward mpox among healthcare workers.**

access to updated guidelines, training opportunities, and reliable information sources that constrain knowledge uniformity across the region [40].

Similarly, the pooled estimate of positive attitudes (53.8%) is higher than the 34.6% reported in a pre-2022 global meta-analysis [14] and aligns with a recent global estimate of 53.0% [15], a shift that may be attributed to increased perceived threat and preparedness efforts after 2022. Our meta-analysis finding is notably higher than that reported in several individual studies from Vietnam [36], Egypt [38], Nigeria [28], and Ethiopia [22], which reported 33.0%, 44.5%, 44.5%, and 49.2% respectively. This variation likely reflects differences in cultural values, beliefs, and social norms, which can significantly shape attitudes toward infectious diseases and their management.

However, our pooled estimate of positive attitudes was lower than those reported in separate studies from Ethiopia [21], Nigeria [26], and Uganda [34], which reported 62.0%, 69.9%, and 80.0% respectively. This discrepancy may be due to the inclusion of studies with a wider range of attitude scores in our synthesis, where the influence of highly positive studies may have been balanced by those with more moderate or lower scores. Such variability underscores the context-dependent nature of attitude formation and highlights the importance of considering local cultural and institutional factors when designing training and communication strategies. The persistent knowledge gaps and variable attitudes among healthcare workers in SSA highlight an urgent need for structured, regionally tailored interventions.

The lack of significant associations in meta-regression suggests that temporal trends and study scale are not primary drivers of the variability in mpox knowledge. Instead, contextual factors such as regional training programs, exposure to outbreaks, and access to updated guidelines may play more influential roles. The high heterogeneity observed across both good knowledge and positive attitudes outcomes underscores the influence of diverse study settings, methodologies, and local healthcare contexts.

Ministries of Health, in collaboration with the WHO and Africa CDC, should develop and disseminate standardized, context-appropriate mpox training modules that address transmission, prevention, diagnosis, and management; integrate mpox education into pre-service and in-service training curricula for all healthcare personnel; leverage digital platforms and mobile health tools to deliver just-in-time updates, guidelines, and refresher training to frontline workers; and implement regular knowledge and attitude assessments to monitor preparedness, identify localized gaps, and evaluate the

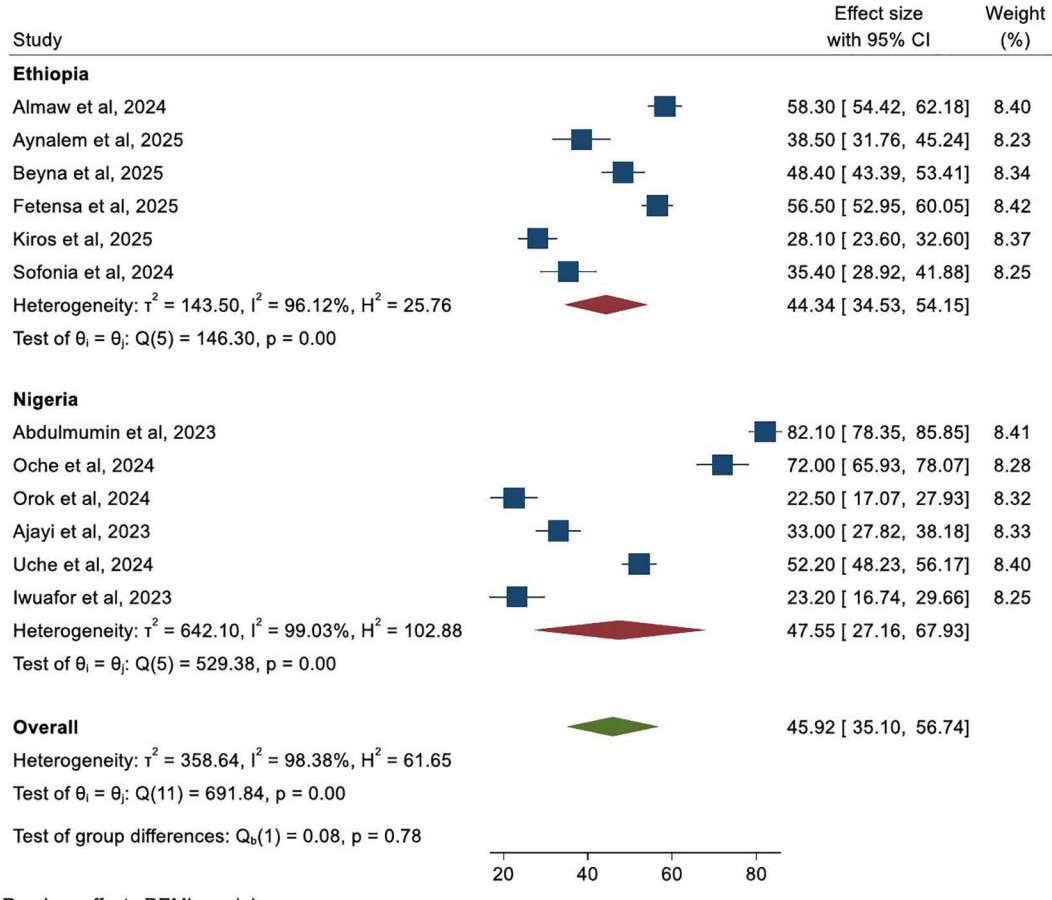

**Fig 5. Forest plot illustrating subgroup analysis comparing mpox knowledge among healthcare workers in Ethiopia and Nigeria.**

impact of training programs. Such targeted efforts are essential to bridge the current knowledge-practice gap and enhance outbreak readiness across SSA.

This review has several limitations, including reliance on cross-sectional studies, potential language bias, and high heterogeneity across included studies, which caution against over-generalization of the findings. Future research should employ longitudinal designs, include non-English publications, and explore the specific contextual, cultural, and institutional factors that influence mpox-related knowledge and attitudes in SSA. Despite these limitations, the present synthesis provides a robust evidence base to guide policy and training initiatives aimed at strengthening mpox preparedness in the region.

## Limitations of the study

While providing valuable insights into healthcare workers knowledge and attitudes toward mpox in Sub-Saharan Africa, this review has several limitations that should be considered when interpreting the findings. The high statistical heterogeneity observed across studies ($I^2 = 98.1\%$ for knowledge, $I^2 = 97.7\%$ for attitudes) indicates substantial variability in study designs, settings, measurement tools and definitions of outcome, and participant characteristics, which may limit the generalizability and comparability of the pooled estimates.

Additionally, the restriction to English-language articles may have introduced language bias, potentially excluding relevant evidence from non-English publications in Sub-Saharan Africa. The cross-sectional design of the included studies

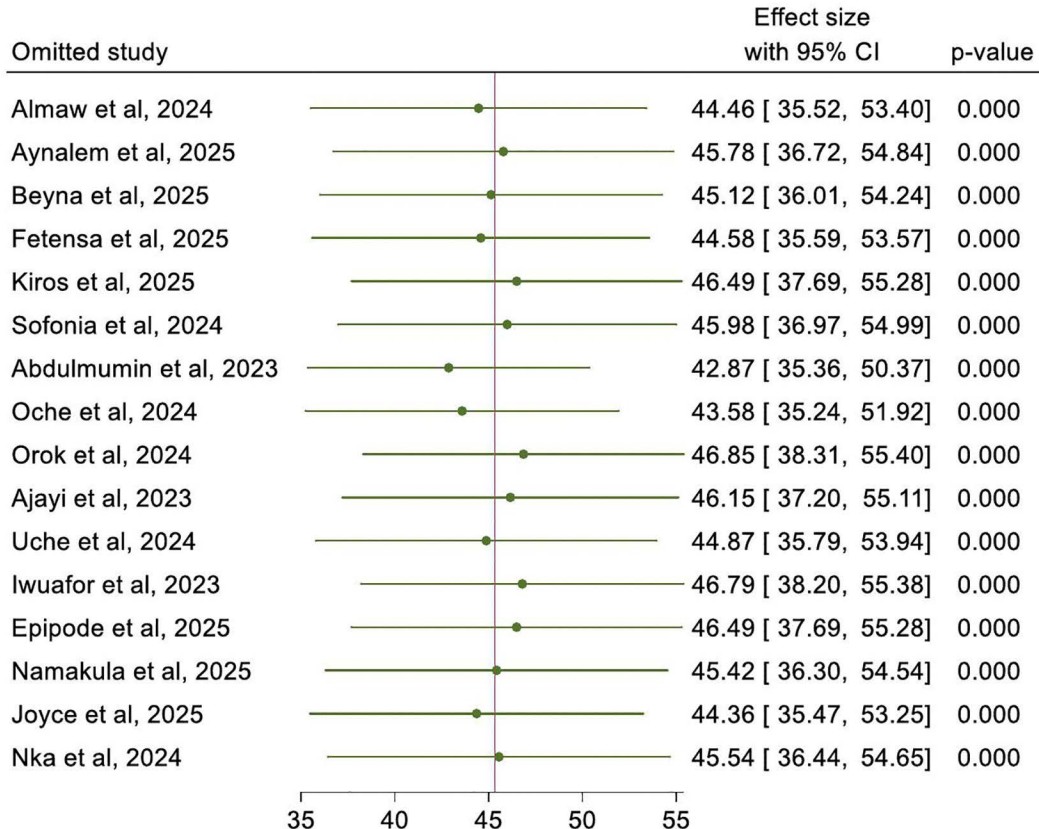

**Fig 6. Sensitivity test for the mpox knowledge prevalence among healthcare workers in Sub-Saharan Africa.**

may be susceptible to confounding variables, and rapidly evolving knowledge about mpox means that the included studies might not fully reflect the most up-to-date information. Variations in questionnaire design, administration, and participant demographics also pose challenges.

Moreover, variability in cultural, socioeconomic, and governmental factors may influence the outcomes. Due to limited data, subgroup analyses were restricted, hindering country-specific conclusions. The small number of attitude studies precluded subgroup analysis and publication bias assessment, limiting robustness. In addition, one primary study (Iwuafor, 2023) was not clearly reported the ethical statements, as result results should be interpreted in cautious manner.

These limitations warrant cautious interpretation and highlight areas for future research. Nevertheless, we employed rigorous methods, including independent study selection, data extraction, and quality assessment with the JBI tool, which strengthens the validity of our findings.

## Conclusion

The good knowledge and positive attitudes toward mpox among healthcare workers in Sub-Saharan Africa were low and unsatisfactory. This meta-analysis reveals that less than half (45.3%) of healthcare workers had good knowledge of mpox, and while more than half of healthcare workers had positive attitudes (53.8%), both measures highlight significant gaps in awareness and preparedness. These findings underscore the urgent need for targeted educational programs, training initiatives, and context-specific interventions to enhance healthcare workers' understanding of mpox transmission,

prevention, and management. These efforts are critical for improving healthcare workers good knowledge and positive attitudes, thereby strengthening mpox control and preparedness in Sub-Saharan Africa. A more informed and prepared healthcare workforce can also improve the early detection of future outbreaks both for mpox and other emerging infectious diseases.

However, these conclusions should be interpreted in light of certain limitations, including the reliance on cross-sectional designs, potential language bias due to English-only inclusion, and heterogeneity across studies in terms of measurement tools and settings. We now recommend that future studies adopt standardized, validated instruments to assess mpox good knowledge and positive attitudes, which would enhance comparability and the robustness of future meta-analyses. Despite these limitations, the results provide a robust evidence base to inform public health strategies aimed at strengthening mpox control and preparedness in Sub-Saharan Africa.

## Supporting information

**S1 File. PRISMA 2020 checklist.**
(DOCX)

**S2 File. Full search strategies or strings.**
(PDF)

**S3 File. Included & excluded studies table.**
(DOCX)

**S4 File. Data extraction excel sheet.**
(XLSX)

**S5 File. Missing data handling.**
(DOCX)

## Author contributions

**Conceptualization:** Melaku Laikemariam.

**Data curation:** Melaku Laikemariam, Alemayehu Molla Wollie, Amare Mebrat Delie, Yihalem Abeje.

**Formal analysis:** Melaku Laikemariam.

**Methodology:** Melaku Laikemariam, Yihalem Abeje.

**Project administration:** Melaku Laikemariam.

**Resources:** Alemayehu Molla Wollie, Amare Mebrat Delie, Yihalem Abeje.

**Software:** Alemayehu Molla Wollie, Amare Mebrat Delie.

**Supervision:** Abebe Yenesew, Abateneh Melkamu, Habtamu Ayele.

**Validation:** Abebe Yenesew, Abateneh Melkamu, Habtamu Ayele.

**Writing – original draft:** Melaku Laikemariam.

**Writing – review & editing:** Melaku Laikemariam.

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
