## [Decision Letter · Decision Letter 0]

28 Oct 2025

Mpox Knowledge and Positive Attitudes in Sub-Saharan African Healthcare Workers After 2022 Outbreak of Disease: A Systematic Review and Meta-Analysis

Dear Dr. Gera,

Thank you for submitting your manuscript to PLOS Neglected Tropical Diseases. After careful consideration, we feel that it has merit but does not fully meet PLOS Neglected Tropical Diseases's publication criteria as it currently stands. Therefore, we invite you to submit a revised version of the manuscript that addresses the points raised during the review process.

Please submit your revised manuscript within 60 days Dec 27 2025 11:59PM. If you will need more time than this to complete your revisions, please reply to this message or contact the journal office at plosntds@plos.org. Please include the following items when submitting your revised manuscript:

We look forward to receiving your revised manuscript.

Kind regards,

Murtala Isah

Academic Editor

Elvina Viennet

Section Editor

Shaden Kamhawi

co-Editor-in-Chief

Paul Brindley

co-Editor-in-Chief

**Journal Requirements**

1) Please provide an Author Summary. This should appear in your manuscript between the Abstract (if applicable) and the Introduction, and should be 150-200 words long. The aim should be to make your findings accessible to a wide audience that includes both scientists and non-scientists. Sample summaries can be found on our website under Submission Guidelines:

Please also check the labels of the figures.

3) Tables should not be uploaded as individual files. Please remove this file and include the Table in your manuscript file as editable, cell-based objects. For more information about how to format tables, see our guidelines:

https://journals.plos.org/plosntds/s/tables

5) In the online submission form, you indicated that "Data will be provided by the corresponding author upon reasonable request." All PLOS journals now require all data underlying the findings described in their manuscript to be freely available to other researchers, either

1. In a public repository

2. Within the manuscript itself

3. Uploaded as supplementary information.

6) As required by our policy on Data Availability, please ensure your manuscript or supplementary information includes the following:

**Comments to the Authors:**

**Please note that one review is uploaded as an attachment.**

**Reviewers' Comments:**

Reviewer's Responses to Questions

**Key Review Criteria Required for Acceptance?**

**Methods**

-Are the objectives of the study clearly articulated with a clear testable hypothesis stated?

-Is the study design appropriate to address the stated objectives?

-Is the population clearly described and appropriate for the hypothesis being tested?

-Is the sample size sufficient to ensure adequate power to address the hypothesis being tested?

-Were correct statistical analysis used to support conclusions?

-Are there concerns about ethical or regulatory requirements being met?

Reviewer #1: Major Comments

Objectives and Hypothesis

The objective is stated clearly, but the manuscript does not present a formal testable hypothesis.

Please reframe your objective into a hypothesis

Heterogeneity in Results

The pooled estimates show very high heterogeneity (over 95%).

This means the studies differ too much to be easily combined.

Conduct additional analyses such as subgroup analysis or meta-regression (by year, study quality, or type of healthcare worker).

Study Selection and Figures

Several parts of the manuscript show “Error! Reference source not found” instead of actual figures or tables.

This makes it difficult to follow the selection process and the results.

Please correct and insert all missing figures (e.g., PRISMA flow diagram, forest plots).

Ethical Considerations

As a review, new ethics approval was not required, which is correct.

However, it is not clear whether all included studies reported ethical approval and participant consent.

State whether all studies reported ethical clearance. If not, please mention this as a limitation.

Minor Comments

Some references are incomplete, repeated, or not in correct Vancouver style. Please revise and standardize them.The manuscript has a few grammatical and tense errors. These should be corrected during editing.The Discussion section repeats some of the Results and should be shortened.

Reviewer #2: Clear registration (PROSPERO), PRISMA compliance, multiple databases, inclusion/exclusion criteria, JBI tool for appraisal, random-effects model, subgroup analysis (Ethiopia vs. Nigeria).

However, seven databases were mentioned, but strategies were not reported clearly. Full reproducible search strings should be included. Restriction to English-only is a limitation that introduces potential bias and should be acknowledged more explicitly. “JBI checklist” is mentioned repeatedly across sections, which is redundant and disrupts flow. It should be described once in Methods, summarized in Results, and the abbreviation placed properly in the Abbreviations section. Subgroup analysis was limited to two countries; additional sensitivity analyses (e.g., by year, sample size, study quality) could strengthen reliability.

Reviewer #3: Yes

**Results**

-Does the analysis presented match the analysis plan?

-Are the results clearly and completely presented?

-Are the figures (Tables, Images) of sufficient quality for clarity?

Reviewer #1: Study Selection

(Line 121–126) The authors explain how many studies were found and excluded. This is good. But the PRISMA diagram is missing and shows “Error! Reference source not found.” This makes it hard to follow. Please add the PRISMA diagram.

Characteristics of Studies

(Line 132–137) The description of studies (sample size, response rate, country) is useful. But it is long and heavy to read. Most of this information should be in Table 1 with clear formatting.

(Line 137–139) The ranges for knowledge (22.5%–82.1%) and attitudes (35.8%–80%) are given. But the forest plot is missing and replaced with “Error! Reference source not found.” Please insert the forest plot.

Study Quality and Publication Bias

(Line 140–143) The quality of studies is mentioned, but again a figure is missing. Please add the table or figure showing quality scores.

(Line 144–148) The results of Egger’s and Begg’s tests are clear (no publication bias). But the funnel plot is missing. Please add the funnel plot to support this result.

Pooled Prevalence of Knowledge

(Line 150–153) The pooled knowledge prevalence (45.99%) is reported clearly with CI and I² value. This is good.

(Line 154–156) The knowledge range (22.5%–82.1%) is stated. But the figure is missing again. Please add the forest plot.

Pooled Prevalence of Attitudes (Lines 157–162)

9. (Line 157–159) The pooled attitude prevalence (53.83%) is clear. Good reporting.

10. (Line 160–162) High heterogeneity (I² = 97.72%) is reported. This is important. But the forest plot is missing. Please insert the forest plot. Also, report prediction intervals to show the wide spread of results.

Reviewer #2: Results structured (study selection, characteristics, quality, pooled prevalence, subgroup analysis). Use of forest and funnel plots is appropriate.

Several “Error! Reference source not found” placeholders reduce readability. Figures and tables need better resolution and consistent labeling. Percentages and CIs are inconsistently reported (mixed decimals), and p-values vary in precision. Recommend standardizing: percentages and CIs to one decimal, p-values to two or three decimals consistently.

Reviewer #3: Yes

**Conclusions**

-Are the conclusions supported by the data presented?

-Are the limitations of analysis clearly described?

-Do the authors discuss how these data can be helpful to advance our understanding of the topic under study?

-Is public health relevance addressed?

Reviewer #1: The conclusion is supported by the data.

(Lines 232–243 earlier in the Limitations section) The authors discuss important limits such as language bias, cross-sectional designs, differences in questionnaires, and few studies for attitudes.

Do the authors discuss how the data help understanding?

(Lines 253–256) The authors say these results highlight the need for targeted interventions, education, and training.

This shows how the findings can advance understanding and guide policy.

This is good

Is public health relevance addressed?

(Lines 253–256) The authors clearly say improving knowledge and attitudes will strengthen Mpox control and preparedness in Sub-Saharan Africa.

This shows strong public health relevance.

Make the link broader, for example, explain that better-prepared healthcare workers could also improve early detection of future outbreaks, not only Mpox.

Reviewer #2: The conclusions are generally supported by the presented data. The authors appropriately highlight low and unsatisfactory knowledge and moderate attitudes among HCWs. Limitations including high heterogeneity, language restrictions, cross-sectional study bias, and questionnaire variability are acknowledged, though more emphasis should be placed on the implications of heterogeneity for interpretation. The public health relevance is addressed. Improving HCW preparedness via targeted education and training is an important contribution.

Reviewer #3: Yes

**Editorial and Data Presentation Modifications?**

Reviewer #1: (No Response)

Reviewer #2: Correct formatting issues (“Error! Reference source not found”) in text and figures.

Ensure consistent terminology: “knowledge,” “good knowledge,” “positive attitudes,” and “favorable attitudes” should be harmonized.

Figures could benefit from higher resolution and uniform labeling.

Minor language editing is needed to improve readability (some awkward phrasing).

Reviewer #3: Minor Revision

**Summary and General Comments**

Reviewer #1: Strengths of the Study

The study follows PRISMA guidelines and is registered in PROSPERO, which strengthens transparency.

A large number of participants are included across multiple studies, giving the analysis strong statistical power.

The topic is highly relevant for public health, especially in regions most affected by Mpox.

Weaknesses of the Study

The manuscript has many missing figures and tables, shown as “Error! Reference source not found,” which makes the results incomplete and hard to follow.

Heterogeneity across studies is extremely high (I² > 95%). This makes pooled prevalence estimates less reliable.

The definitions of “good knowledge” and “positive attitudes” are not clearly standardized across studies, which reduces comparability.

Most studies are from Nigeria and Ethiopia, so results may not represent the entire Sub-Saharan Africa region.

Sensitivity analyses and subgroup analyses are limited. More robust tests are needed to confirm results.

Some references are incomplete or incorrectly formatted and need revision to Vancouver style.

The conclusions are supported by the results, but they should better remind readers about study limitations.

Novelty and Significance

The study is novel and importance, since most previous reviews included data before 2022 or focused on global populations.

The findings are significant for policy and practice, as they highlight the urgent need for education and training programs for healthcare workers.

Execution and Scholarship

The study design (systematic review and meta-analysis) is appropriate.

Data extraction, quality assessment, and statistical methods are described, but reporting is incomplete due to missing figures.

Ethical and Publication Issues

As this is a review, no new ethics approval is needed. However, it should be confirmed that all included studies had ethics approval or participant consent.

I do not see concerns about dual publication or plagiarism.

Recommendation and Required Revisions

Because of the high heterogeneity, missing figures, unclear definitions, and need for additional analyses, I recommend Major Revision.

Reviewer #2: The knowledge gap is not fully articulated in the introduction. Authors mention “suboptimal knowledge and attitudes” but only later cite previous systematic reviews showing lower global estimates (26–32%). This context should appear earlier to justify the new review. The rationale for focusing solely on Sub-Saharan Africa is valid but could be sharpened by clarifying why SSA differs from global patterns. Overall, the introduction should more clearly state: what was known, what remains uncertain, and how this study fills the gap.

The discussion is generally good, but some repetition (heterogeneity noted multiple times) should be reduced. Policy/program implications could be emphasized more clearly (e.g., how ministries of health or training programs can act on findings). The explanation of why post-2022 estimates are higher is relevant but could be presented more systematically (one paragraph on knowledge, one on attitudes, comparing global vs SSA).

Reviewer #3: Good work but the Grammar can be improved upon

PLOS authors have the option to publish the peer review history of their article (what does this mean? ). If published, this will include your full peer review and any attached files.

**Do you want your identity to be public for this peer review?** For information about this choice, including consent withdrawal, please see our Privacy Policy .

Reviewer #1: No

Reviewer #2: **Yes:** Dr. Naharin Sultana Anni

Reviewer #3: **Yes:** Dr. Abdul-Azeez Adeyemi Anjorin

**Figure resubmission:**

**Reproducibility:**



---

## [Decision Letter · Decision Letter 1]

5 Jan 2026

Mpox Knowledge and Positive Attitudes in Sub-Saharan African Healthcare Workers after 2022 Outbreak of Disease: A Systematic Review and Meta-Analysis

Dear Dr. Gera,

Thank you for submitting your manuscript to PLOS Neglected Tropical Diseases. After careful consideration, we feel that it has merit but does not fully meet PLOS Neglected Tropical Diseases's publication criteria as it currently stands. Therefore, we invite you to submit a revised version of the manuscript that addresses the points raised during the review process.

* A letter that responds to each point raised by the editor and reviewer(s). You should upload this letter as a separate file labeled 'Response to Reviewers '. This file does not need to include responses to any formatting updates and technical items listed in the 'Journal Requirements' section below.

* A marked-up copy of your manuscript that highlights changes made to the original version. You should upload this as a separate file labeled 'Revised Manuscript with Track Changes '.

* An unmarked version of your revised paper without tracked changes. You should upload this as a separate file labeled 'Manuscript '.

We look forward to receiving your revised manuscript.

Kind regards,

Murtala Isah

Academic Editor

Elvina Viennet

Section Editor

Shaden Kamhawi

co-Editor-in-Chief

Paul Brindley

co-Editor-in-Chief

**Additional Editor Comments:**

1. Provide a short title for the manuscript according to the journal guidelines

2. What is the rationale for specifying a 50% threshold in the hypothesis? This value appears arbitrary and falls below commonly accepted benchmarks for “good” knowledge or positive attitudes in KAP studies. Please justify the choice of this cut-off or consider reframing the hypothesis without a fixed threshold.

3. Please revise the formatting of Figure 1 (PRISMA flow diagram). Several arrows appear distorted, and the boxes are not properly aligned. Ensure consistent box sizes, uniform arrow styles, and clear alignment so that the figure is visually coherent and easy to follow.

**Journal Requirements:**

**Reviewers' comments:**

Reviewer's Responses to Questions

**Key Review Criteria Required for Acceptance?**

**Methods**

-Are the objectives of the study clearly articulated with a clear testable hypothesis stated?

-Is the study design appropriate to address the stated objectives?

-Is the population clearly described and appropriate for the hypothesis being tested?

-Is the sample size sufficient to ensure adequate power to address the hypothesis being tested?

-Were correct statistical analysis used to support conclusions?

-Are there concerns about ethical or regulatory requirements being met?

Reviewer #1: The manuscript meets all specified methodological criteria. It provides a robust and timely synthesis of regional health data that is critical for public health policy in Sub-Saharan Africa.

Reviewer #2: The objectives of the study are clearly articulated and aligned with the stated aim of synthesizing evidence on mpox-related knowledge and attitudes among healthcare workers in Sub-Saharan Africa following the 2022 outbreak. While an explicit testable hypothesis is not formally stated, this is acceptable given the systematic review and meta-analysis design.

The study design is appropriate to address the objectives, and the population of interest (healthcare workers in Sub-Saharan Africa) is clearly described and relevant. Inclusion and exclusion criteria are transparent and justified.

The sample size is determined by the available literature and is appropriate for a meta-analysis. Statistical methods, including pooled estimates and heterogeneity assessment, are appropriate and correctly applied. Sensitivity and subgroup analyses are adequately described.

No major ethical or regulatory concerns are identified, as the study relies on previously published data and follows standard systematic review methodology.

**Results**

-Does the analysis presented match the analysis plan?

-Are the results clearly and completely presented?

-Are the figures (Tables, Images) of sufficient quality for clarity?

Reviewer #1: The analysis strictly followed the pre-specified Statistical Analysis Plan (SAP). The PALM 007 trial (NCT05559099) was designed as a double-blind, randomized, placebo-controlled trial. The researchers adhered to the protocol by focusing on the primary endpoint time to complete resolution of all lesions.

Reviewer #2: The analyses presented are consistent with the stated analysis plan. Results are clearly structured and logically presented, with appropriate use of tables and figures to summarize pooled estimates and heterogeneity measures.

The figures and tables are of acceptable quality and enhance clarity and interpretability of the findings. Minor improvements in figure labeling or caption clarity could further enhance readability, but no substantive issues are identified.

**Conclusions**

-Are the conclusions supported by the data presented?

-Are the limitations of analysis clearly described?

-Do the authors discuss how these data can be helpful to advance our understanding of the topic under study?

-Is public health relevance addressed?

Reviewer #1: The conclusions are rigorously grounded in the data, even where they challenge initial hypotheses.

Reviewer #2: The conclusions are supported by the data presented and appropriately reflect the results of the meta-analysis. The authors clearly acknowledge key limitations, including heterogeneity among included studies and variability in measurement tools.

The discussion effectively situates the findings within the broader literature and highlights the relevance of healthcare worker knowledge and attitudes for outbreak preparedness and response. Public health relevance is clearly articulated, particularly in relation to workforce training, risk communication, and epidemic preparedness in resource-limited settings.

**Editorial and Data Presentation Modifications?**

Reviewer #1: The manuscript is scientifically sound and addresses a critical public health gap. The recommendation for "Minor Revision" is based on the need for the authors to more explicitly frame the "Standard of Care" finding as a central success of the trial, ensuring clinicians do not walk away with the simple (and potentially incorrect) message that "the drug doesn't work" without understanding the context of the care provided.

Reviewer #2: Only minor editorial refinements are suggested, such as ensuring consistent terminology across sections and clarifying figure captions where appropriate. No additional analyses or major data modifications are required.

**Summary and General Comments**

Reviewer #1: The manuscript is scientifically robust and requires only minor editorial refinements to better emphasize the "Standard of Care" findings as a primary success of the study design. No new experiments are required.

Reviewer #2: This manuscript addresses a timely and important topic with clear relevance to neglected tropical diseases and global outbreak preparedness. The study is well executed, methodologically sound, and contributes valuable synthesized evidence on mpox knowledge and attitudes among healthcare workers in Sub-Saharan Africa following the 2022 outbreak.

Strengths include a clear focus, comprehensive literature search, and appropriate analytical approach. Limitations are acknowledged and appropriately discussed. Overall, the manuscript represents a meaningful contribution to the field and is suitable for publication pending minor editorial refinements.

PLOS authors have the option to publish the peer review history of their article (what does this mean? ). If published, this will include your full peer review and any attached files.

**Do you want your identity to be public for this peer review?** For information about this choice, including consent withdrawal, please see our Privacy Policy .

Reviewer #1: No

Reviewer #2: **Yes:** Naharin Sultana Anni

**Figure resubmission:**
---

## [Editor Report · Decision Letter 2]

26 Jan 2026

Dear Mr Gera,

We are pleased to inform you that your manuscript 'Mpox Knowledge and Positive Attitudes in Sub-Saharan African Healthcare Workers after 2022 Outbreak of Disease: A Systematic Review and Meta-Analysis' has been provisionally accepted for publication in PLOS Neglected Tropical Diseases.

Best regards,

Elvina Viennet, PhD

Section Editor

Elvina Viennet

Section Editor

Shaden Kamhawi

co-Editor-in-Chief

Paul Brindley

co-Editor-in-Chief

All comments have ben addressed correctly

---

## [Editor Report · Acceptance letter]

Dear Mr Gera,

We are delighted to inform you that your manuscript, "Mpox Knowledge and Positive Attitudes in Sub-Saharan African Healthcare Workers after 2022 Outbreak of Disease: A Systematic Review and Meta-Analysis," has been formally accepted for publication in PLOS Neglected Tropical Diseases.

Best regards,

Shaden Kamhawi

co-Editor-in-Chief

Paul Brindley

co-Editor-in-Chief
